# Type 1 or Type 2 Myocardial Infarction in Patients with a History of Coronary Artery Disease: Data from the Emergency Department

**DOI:** 10.3390/jcm8122100

**Published:** 2019-12-02

**Authors:** Alain Putot, Mélanie Jeanmichel, Frédéric Chagué, Aurélie Avondo, Patrick Ray, Patrick Manckoundia, Marianne Zeller, Yves Cottin

**Affiliations:** 1Geriatrics Internal Medicine Department, University Hospital of Dijon Bourgogne, 21079 Dijon CEDEX, France; 2Physiopathologie et Epidémiologie Cérébro-Cardiovasculaires (PEC2), EA 7460, University of Burgundy and Franche Comté, 21079 Dijon CEDEX, France; 3Cardiology Department, University Hospital of Dijon Bourgogne, 21079 Dijon CEDEX, France; 4Emergency Department, University Hospital of Dijon Bourgogne, 21079 Dijon CEDEX, France

**Keywords:** type 2 myocardial infarction, coronary artery disease, myocardial infarction, hospital mortality

## Abstract

A type 2 myocardial infarction (T2MI) is the result of an imbalance between oxygen supply and demand, without acute atherothrombosis. T2MI is frequent in emergency departments (ED), but has not been extensively evaluated in patients with previously known coronary artery disease (CAD). Our study assessed the incidence and characteristics of T2MI compared to type 1 (T1MI) in CAD patients admitted to an ED. Among 33,669 consecutive patients admitted to the ED, 2830 patients with T1MI or T2MI were systematically included after prospective adjudication by the attending clinician according to the universal definition. Among them, 619 (22%) patients had a history of CAD. Using multivariable analysis, CAD history was found to be an independent predictive factor of T2MI versus T1MI (odds ratio (95% confidence interval) = 1.38 (1.08–1.77), *p* = 0.01). Among CAD patients, those with T2MI (*n* = 254) were older (median age: 82 vs. 72 years, *p* < 0.001), and had more frequent comorbidities and more frequent three-vessel disease at the coronary angiography (56% vs. 43%, *p* = 0.015). Percutaneous coronary intervention was by far less frequent after T2MI than after T1MI (28% vs. 67%, *p* < 0.001), and in-hospital mortality was twice as high in T2MI (15% vs. 7% for T1MI, *p* < 0.001). Among biomarkers, the C reactive protein (CRP)/troponin Ic ratio predicted T2MI remarkably well (C-statistic (95% confidence interval) = 0.84 (0.81–0.87, *p* < 0.001). In a large unselected cohort of MI patients in the ED, a quarter of patients had previous CAD, which was associated with a 40% higher risk of T2MI. CRP/troponin ratios could be used to help distinguish T2MI from T1MI.

## 1. Introduction

In 2018, a new universal definition of myocardial infarction (MI) further clarified the initial classification of MI [1]. A type 1 MI (T1MI) is an acute atherothrombotic coronary event resulting in the formation of an intra-luminal thrombus (plaque rupture, ulceration, erosion, or coronary dissection). A type 2 MI (T2MI) has been described as myocardial necrosis occurring in the context of an imbalance between oxygen supply and demand in the absence of an atherothrombotic coronary event, with or without underlying coronary artery disease (CAD). A recent study in our hospital showed that 21% of the patients admitted to the emergency department (ED) with elevated troponin had T2MI [2]. With the ageing of the population, the incidence of T2MI is expected to increase [3]. However, there is currently no solid data to conclude that interventions, such as a coronary angiography or percutaneous coronary intervention (PCI), are beneficial in T2MI. In the absence of evidence-bases therapies, T2MI management relies on correcting the underlying ischemic imbalance [1]. Although the concept of T2MI is increasingly adopted by clinicians, and despite recent coding by the International Classification of Diseases [4], the absence of diagnostic criteria has led to a heterogeneous incidence in the literature: the reports of T2MI range from 2 to 64% of patients with myocardial necrosis [5].

Several authors [6] have suggested limiting the definition of T2MI to a more homogeneous group of patients with “stable pre-existing” CAD decompensated by an acute non-atherothrombotic phenomenon, thus reflecting the criteria of the 2007 universal definition. Among MI patients, those with a CAD history constitute a specific group for whom secondary preventive strategies have failed. Whether these patients more frequently have T1MI or T2MI is to date unknown. Moreover, as acute care management largely differs, there is a need for easy-to-use distinctive markers for rapid discrimination between the two pathophysiological types of MI. 

In this context, using a large database of patients admitted to the ED for MI, the objectives of our study were: (1) to define whether a history of CAD more likely predisposes a patient to T1MI or T2MI, and (2) to determine the distinctive characteristics of T2MI versus T1MI in the specific group of patients with previously known CAD.

## 2. Methods

### 2.1. Study Population

Our observational study prospectively included all consecutive patients over 18 years of age hospitalized for T1MI or T2MI in the emergency department or in the cardiology department of the University Hospital of Dijon, France, between 1 January 2014 and 31 December 2016. All consecutive patients with elevated troponin Ic (≥0.10 µg/L) in relation to a type 1 or type 2 MI were included, i.e., in the presence of symptoms or signs of myocardial ischemia (typical chest pain and/or electrocardiogram (ECG) changes) according to the criteria of the current universal definition of MI [7]. Patients with type 3, 4, or 5 MI or myocardial injury or who were admitted for non-emergency interventions or with missing data (0.7%) were excluded. Myocardial injury was defined as the presence of an elevated troponin level without any evidence of ischemia [7].

The present study complied with the Declaration of Helsinki and was approved by the Ethics Committee of the Dijon University Hospital.

### 2.2. Diagnostic Classification

The type of MI was prospectively adjudicated by the attending clinician based on clinical and angiographic criteria according to the third universal definition [7]. The validation of the type of MI was performed retrospectively by a cardiologist participating in the study. In the event of a discrepancy, the case was analyzed by another cardiologist. Patients classified as having T1MI had myocardial necrosis secondary to an atherosclerotic plaque rupture, ulceration, cracking, erosion, or dissection with intra-luminal thrombus and obstruction of myocardial blood flow. Patients classified as having T2MI had myocardial necrosis resulting from an increase in oxygen demand (e.g., tachyarrhythmia or hypertrophy) and/or a decrease in myocardial blood flow (e.g., hypotension, hypoxia, anemia, or infection) [8,9] occurring in the absence of an acute plaque rupture or coronary thrombosis. 

A history of CAD was defined by the presence of at least one of the following: a history of myocardial infarction or PCI, a history of coronary artery bypass grafting (CABG), a history of stable angina, or a positive ischemia test.

### 2.3. Data Collection

The following parameters were collected from individual medical records: the patient’s age and gender, cardiovascular risk factors (dyslipidemia, insulin-requiring or non-insulin-requiring type 1 or 2 diabetes, overweight or obesity, current or past smoking, family history of CAD, hypertension), the presence of documented CAD (history of angina, CABG, PCI, MI, unstable angina, positive ischemia test) and the time to entry into CAD, medical history (heart failure (HF), chronic kidney disease (CKD), stroke, peripheral arteriopathy), chronic treatments: antiplatelet agents (aspirin, clopidogrel, ticagrelor, prasugrel) and/or anticoagulants (heparin, anti-vitamin K or direct oral anticoagulants), clinical presentation (heart rate, blood pressure), and ECG data at admission (presence of ST-segment-elevation myocardial infarction (STEMI)), biological parameters at admission (C reactive protein (CRP), creatinine, hemoglobin), coronary angiography data when available: number of significant lesions, defined by the presence of stenosis ≥50% and the coronary lesion severity score (SYNTAX) [10]. The creatinine thresholds for defining acute kidney injury on admission were: >85 µmol/L for women and >104 µmol/L for men. Cardiac troponin I was measured with the Siemens Dimension Vista^®^ method every 8 h to determine the peak. The minimum detection threshold was 0.015 ng/mL. The test result was considered positive when the troponin level was ≥0.10 µg/L. Data were obtained via manual extraction from medical records for clinical data and electronically extracted for biological parameters. An independent validation was retrospectively performed for the clinical characteristics. 

### 2.4. Statistical Analyses

Patients were divided into two groups based on whether there was documented prior CAD. Dichotomous variables were expressed as numbers (%) and continuous variables as means ± SD or medians (interquartile range). The normality of the continuous data was tested using the Kolmogorov–Smirnov test. Mann–Whitney (two groups) tests were used to compare continuous data. The dichotomous variables were compared with the chi-squared test. 

A multivariate logistic regression model was used for the total population with MI (*n* = 2830) in order to estimate the independent risk factors for T2MI. Variables that were significantly associated in the univariate analysis with a 5% threshold were introduced into the multivariate model: age, female sex, obesity, current smoking, hypercholesterolemia, diabetes, history of CAD, hypertension, family history of CAD, chronic kidney disease, creatinine, troponin peak, CRP > 3 mg/L, and STEMI. The interactions between variables with a potential relationship (i.e., age × CRP > 3mg/L; troponin peak × STEMI; hypertension × chronic kidney disease; history of CAD × STEMI: sex × STEMI; sex × smoking) were tested and added to the multivariate model if significant. Multivariate logistic regression analyses were also conducted to determine the factors associated with T2MI in CAD patients. The variables tested in the univariate analysis were age, female sex, obesity, current smoking, hypercholesterolemia, diabetes, hypertension, family history of CAD, history of heart failure, chronic kidney disease, creatinine, troponin peak, CRP > 3 mg/L, and STEMI. To compare the accuracy of biomarkers and to identify the best cut-off values for T2MI prediction, we constructed receiver operating curves (ROC) and determined the area under the curves (AUCs), as well as the sensitivity, specificity, accuracy, and positive and negative likelihood ratios for each parameter. A *p*-value < 0.05 was considered significant. All the analyses were done with SPSS software version 12.0.1 (IBM Inc., Armonk, NY, USA).

## 3. Results

### 3.1. All MI Patients

Out of the 33,669 patients who were tested for cardiac troponin during the study period, 4283 had positive results. Of these patients, 2830 were diagnosed with T1MI (69.2%) or T2MI (30.8%) (Figure 1). Among patients with no history of CAD, most were diagnosed with T1MI (72%) and just over a quarter (28%) with T2MI. 

From the multivariate analysis of the total population (*n* = 2830), it was found that a history of CAD was a risk factor for T2MI (OR (95% CI) = 1.38 (1.08–1.77), *p* = 0.01), even after adjustments for confounding factors (age, female gender, CRP > 3 mg/L, creatinine (Table 1)). Significant interactions were tested in the multivariate model (age × CRP > 3 mg/L, troponin peak × STEMI, hypertension × chronic kidney disease, history of CAD × STEMI, sex × STEMI; sex × smoking), without changing the results of the model.

### 3.2. Patients with CAD Disease

#### 3.2.1. Risk Factors and History

Among patients with previous CAD (Table 2), 59% were diagnosed with T1MI and 41% with T2MI. Compared with T1MI patients, patients with T2MI were about 10 years older (*p* < 0.001) and there was a higher proportion of women and patients with hypertension (*p* = 0.03). Conversely, patients with T2MI were less likely to smoke and had less family history of CAD. The rates of diabetes and dyslipidemia were similar in both groups. In terms of history, T2MI patients had more frequent heart failure (*p* < 0.001), stroke (*p* = 0.002), and peripheral arteriopathy and chronic kidney disease (*p* = 0.002). In addition, they were treated more often with anticoagulants and treated less often with anti-platelet agents.

#### 3.2.2. Presentation at Admission

At admission, T2MI patients had higher creatinine and CRP levels than T1MI patients (Table 2). T2MI patients had comparatively lower hemoglobin levels (13.8 g/dL for T1MI vs. 12.1 g/dL, *p* < 0.001), whereas T1MI patients had a troponin peak approximately 8 times higher (8.2 vs. 0.8 µg/L for T2MI, *p* < 0.001). 

Figure 2 displays the ROC curves for the predictive biomarkers of T2MI versus T1MI in patients with a history of coronary artery disease. The CRP/troponin I ratio had the best predictive values with an AUC of 0.84 (95% confidence interval: 0.81–0.87).

#### 3.2.3. Coronary Angiography Data

Less than half (41%) of patients hospitalized in the cardiac ICU for a T2MI had a coronary angiography procedure (vs. 96% for T1MI) (Table 2), and T2MI patients had a significantly higher rate of three-vessel disease than T1MI patients (56% vs. 43%, *p* < 0.001). However, the SYNTAX score, indicative of the severity of CAD, was similar in both groups. Among patients who had coronary angiography, only 28% of T2MI cases had a PCI, compared with 67% of T1MI patients. Few patients (4% and 8%, respectively) had coronary artery bypass surgery for revascularization and the difference was not significant. 

#### 3.2.4. Overall In-Hospital and Cardiovascular Mortality

In-hospital mortality was twice as high among T2MI patients (15% vs. 6.6% for T1MI, *p* < 0.001) (Table 2). However, the difference in cardiovascular deaths was not significant: 6.3% for T1MI and 9.4% for T2MI (*p* = 0.1). 

#### 3.2.5. Factors Associated with the Occurrence of T2MI

In patients with a history of CAD, the factors associated with T2MI were acute heart failure (OR = 2.98 (1.73–5.14), *p* < 0.001) and CRP > 3 mg/L (OR = 3.53 (2.17–5.75), *p* < 0.001) (Table 3). 

## 4. Discussion

T2MI is increasingly recognized as a frequent condition in patients admitted to the ED. However, differentiation with T1MI is difficult in practice, although required for optimized investigations and management. In addition, it is not yet known whether MI patients with a history of CAD are more at risk of T1MI or T2MI. The identification of distinctive characteristics for T2MI could be used to minimize unnecessary invasive procedures; this is particularly relevant for frail, elderly individuals presenting with T2MI for whom the benefits of coronary angioplasty have not been proven. Toward this purpose, this study covered a large exhaustive population of MI patients, with systematic adjudication between the types of MI (type 1 or type 2). Myocardial injury without necrosis was excluded, as outlined in the new universal definition of MI [1]. 

The main results are as follows: T2MI was common in an unselected population (31% of MI in our cohort), especially among patients with a known history of CAD (41%).The existence of known underlying CAD increased the probability of having T2MI versus T1MI by 40%.In MI patients with a history of CAD, a high CRP/troponin ratio predicted T2MI remarkably well.

### 4.1. CAD Predisposes a Patient to T2MI More Than T1MI

To our knowledge, this study is the first to identify CAD as an independent risk factor for T2MI compared with T1MI. However, several studies have already concluded that a history of CAD is more frequent among T2MI than T1MI patients [11,12,13,14,15,16,17,18,19]. The overall prevalence of proven underlying CAD is of particular interest in T2MI, but reports are highly variable, ranging from 7% to 53% of MI [11,12,13,14,15]. Indeed, an angiographic examination and/or ischemia tests are not routinely performed in patients admitted for T2MI, independent of whether CAD is known, thus leading to a selection bias. In Arora’s work, for example, only 26% of patients in the T2MI group had a coronary angiography compared with 79% in the T1MI group [14]. Because T2MI results from an imbalance in the myocardial oxygen supply, the severity of the underlying CAD is a major risk factor for T2MI. In our study, T2MI patients were more likely to have three-vessel disease than T1MI patients (56 vs. 43%, *p* = 0.015). Our data agree with Gaggin, who found that 41% of patients in the T2MI group had three-vessel disease compared with 27% of patients in the T1MI group [15].

The prognostic importance of underlying CAD is a key issue for the management of T2MI. In the study by Chapman et al. [16], underlying CAD was an independent predictor of major adverse cardiac events at 5 years in patients with T2MI (hazard ratio (95% CI) = 1.71 (1.31–2.24), *p* < 0.001). Yet, in a recent study, CAD was not independently associated with higher in-hospital mortality [2]. To our knowledge, there are currently no specific data for T2MI on the efficacy of validated therapeutics in the secondary prevention of CAD. If this were demonstrated, systematic screening for CAD after T2MI would be justified. 

### 4.2. Differentiation between T1MI and T2MI in CAD Patients

Distinguishing between T1MI and T2MI is difficult in practice. Recent literature suggests that this type of distinction should be clinically based [17]. However, even if the clinical presentation of T2MI somewhat differs, with less chest pain and more acute heart failure at admission [18,19], these elements remain unspecific. The diagnosis of T2MI is often presumptive, as even in acute situations of oxygen imbalance, the existence of a coexisting atherothrombotic event cannot be excluded. Furthermore, even when a coronary angiography is performed, this invasive procedure is not 100% sensitive because plaque disruption and intracoronary thrombus is not perfectly exclusive to T1MI [17]. 

In patients with known CAD, two independent factors were associated with the occurrence of T2MI after disease onset: heart failure and inflammation. These factors probably indirectly reflect the main triggers of T2MI, which are sepsis and heart failure [19]. Elevated inflammatory markers were an independent predictor of T2MI, as in the study by Stein et al. [9]. One explanation could be the frequency of infections, particularly respiratory infections, in the pathogenesis of T2MI [20].

Re-infarction and MI recurrence are increasingly common phenomena in patients with previously diagnosed CAD, and they remain a major cause of morbidity and mortality [21,22]. As CAD is responsible for a decrease in myocardial oxygen supply, patients with a history of CA have a higher risk of T2MI [16]. However, CAD also highly predisposes patients to T1MI [22], for which PCI radically changes the prognosis. Interestingly, when compared with T1MI, T2MI was associated with a two-fold increase of in-hospital mortality, but no significant difference in cardiovascular deaths. These results suggest an excess risk of non-cardiovascular events in the T2MI population, who were 10 years older than T1MI patients and for whom the iatrogenic risk of invasive procedures should be carefully weighed. Conversely, older patients with T1MI should not be excluded from PCI, as PCI has been associated with substantial improvements in health-related quality of life, similar to those in younger patients [23]. In this context, the predictive value of the CRP/troponin ratio is of particular interest in the ED, as it is an easy-to-use tool that could help the physician distinguish T2MI from T1MI. With a cut-off of 17.5 × 10^3^, we found that the CRP/troponin ratio had a specificity of 90% for T2MI versus T1MI, which could help the choice of not performing immediate PCI. Troponin has already been highlighted as a useful biomarker for the distinction between T1MI and T2MI in patients presenting to the ED, especially when considering the delta of concentrations at admission and after 4 h [24,25], but when considered alone, poorly discriminates T2MI from TIMI [26]. Nowak et al. assessed the predictive value of N-terminal-pro brain natriuretic peptide (NT-Pro BNP)/troponin ratio for the differentiation of 25 T1MI cases from 18 T2MI cases in the ED [27]. Although the conclusions were limited by the low number of patients, the area under the receiver operating curve (AUROC) for this ratio was encouraging (AUROC (95% CI) = 0.76 (0.61–0.92)). However, our results highlight that the CRP/troponin ratio was more powerful for T1MI/T2MI differentiation (AUROC (95% CI) = 0.84 (0.81–0.87)). In addition, the cost of CRP testing is lower than for NT-Pro BNP, and this type of sampling is part of usual practice in the ED. These results remain to be confirmed in other series.

### 4.3. Limitations

Our study has several limitations. The study population was identified based on a high concentration of cardiac troponin I measured using a standard test with a diagnostic threshold of 0.10 μg/L. The incidence of MI may have been higher if a more sensitive test was used (i.e., high-sensitivity troponin). Although MI was systematically diagnosed by an experienced clinician and then reassessed by two independent cardiologists with excellent intra-observer agreement, a risk of misclassification remains, particularly for T2MI. The monocentric nature of the study also limits the extrapolation of our results. These results concern a French, predominantly Caucasian population, and thus external validity could be limited in other populations. However, the systematic inclusion of all patients with elevated troponin levels admitted to an emergency department, regardless of the reason for hospitalization, allowed us to recruit as many cases of T2MI as possible. This made it possible to obtain a good overall picture of the affected population, and the most vulnerable elderly patients were frequently excluded. In contrast, coronary angiography data were only available for the patients hospitalized to the cardiology department, resulting in a selection bias. Finally, the usefulness of the CRP/troponin ratio was evaluated in patients with a previous diagnosis of CAD, which limited the overall efficacy of this diagnostic tool for undiagnosed patients at presentation. 

## 5. Conclusions

Our large series of unselected patients admitted to hospital for MI showed that T2MI was frequent, especially in patients with a known history of CAD. The existence of underlying CAD was an independent predictor of T2MI when compared with T1MI. We also found that the CRP/troponin ratio could be of great help to distinguish between T1MI and T2MI in patients with a history of CAD. Further work is needed to confirm these findings and evaluate treatment strategies in CAD patients diagnosed with T2MI. 

## Figures and Tables

**Figure 1 jcm-08-02100-f001:**
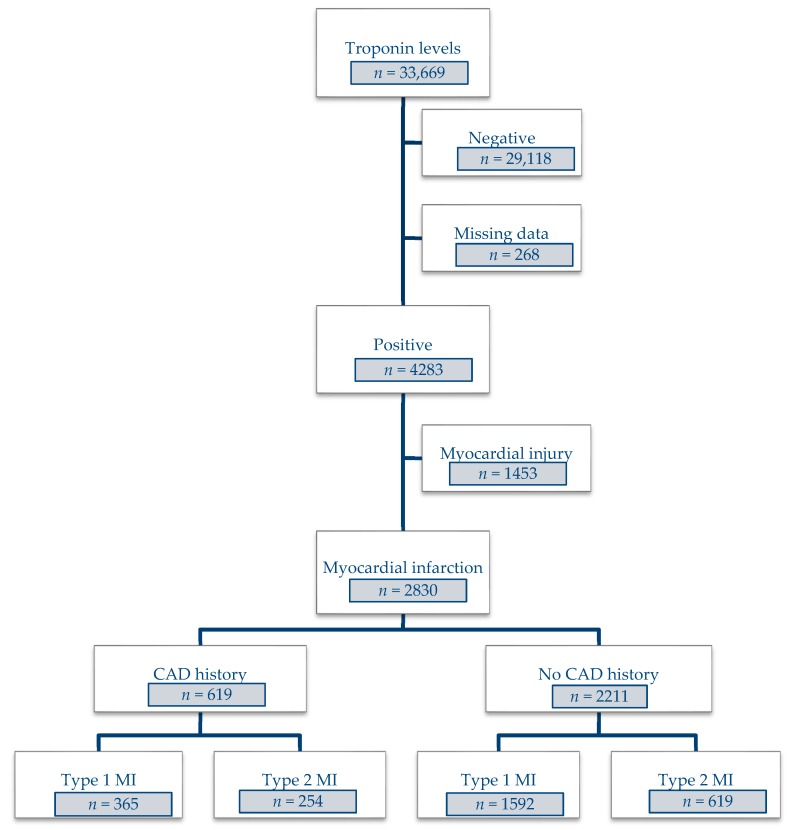
Flow diagram.

**Figure 2 jcm-08-02100-f002:**
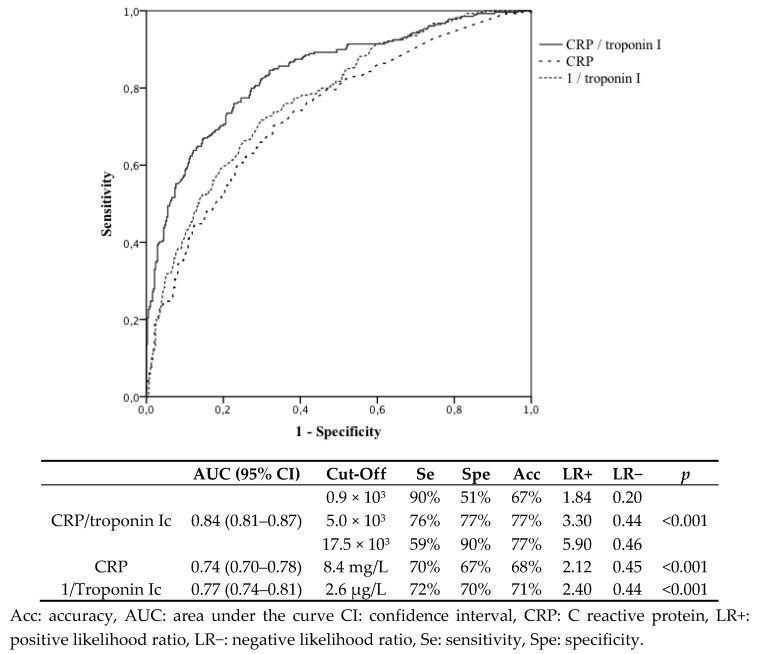
Receiver operating curves of predictive biomarkers for type 2 versus type 1 myocardial infarction in patients with a history of coronary artery disease.

**Table 1 jcm-08-02100-t001:** Logistic regression analysis of the factors associated with T2MI in the whole cohort (*n* = 2830).

	Univariate		Multivariate	
	OR (95% CI)	*p*	OR (95% CI)	*p*
Age	1.06 (1.05–1.07)	<0.001	1.02 (1.01–1.03)	<0.001
Female	2.78 (2.36–3.28)	<0.001	1.64 (1.30–2.06)	<0.001
Obesity	0.39 (0.30–0.50)	<0.001	0.42 (0.31–0.57)	<0.001
Current smoking	0.26 (0.22–0.31)	<0.001	0.48 (0.38–0.61)	<0.001
Family history of CAD	0.33 (0.26–0.42)	<0.001	0.47 (0.35–0.63)	<0.001
Hypertension	1.95 (1.65–2.32)	<0.001	1.05 (0.83–1.32)	0.7
History of CAD	1.79 (11.49–2.15)	<0.001	1.38 (1.08–1.77)	0.010
Chronic kidney disease	2.71 (2.03–3.60)	<0.001	1.45 (0.99–2.12)	0.06
Troponin Ic	0.97 (0.97–0.98)	<0.001	0.99 (0.99–0.99)	<0.001
CRP > 3 mg/L	3.44 (2.79–4.24)	<0.001	2.76 (2.16–3.53)	<0.001
Renal failure at admission	2.60 (2.20–3.08)	<0.001	1.34 (1.05–1.70)	0.02
STEMI	0.11 (0.09–0.13)	<0.001	0.20 (0.15–0.26)	<0.001

CAD: coronary artery disease; CRP: C reactive protein, CI: confidence interval, OR: odds ratio, STEMI: ST-segment elevation myocardial infarction.

**Table 2 jcm-08-02100-t002:** Characteristics of patients with previous coronary artery disease (*n* = 619, *n* (%) or median (IQR)).

	T1MI (*n* = 365)	T2MI (*n* = 254)	*p*
**Risk factors**			
Age, years	72 (62–81)	82 (71–87)	<0.001
Female	84 (23%)	98 (39%)	<0.001
Obesity	97 (27%)	27 (11%)	<0.001
Hypercholesterolemia	259 (71%)	169 (67%)	0.2
Hypertension	276 (76%)	211 (83%)	0.03
Diabetes	138 (38%)	99 (39%)	0.8
Current smoking	243 (67%)	101 (40%)	<0.001
Family history of CAD	108 (30%)	37 (15%)	<0.001
Heart failure	36 (10%)	78 (31%)	<0.001
Stroke	40 (11%)	50 (20%)	0.002
Peripheral arteriopathy	54 (15%)	55 (22%)	0.03
Chronic kidney disease	45 (12%)	55 (22%)	0.002
**Usual treatments**			
Anti-platelet	301 (83%)	189 (74%)	0.01
Anticoagulant	55 (15%)	69 (27%)	<0.001
**Clinical data**			
HR, bpm	76 (66–92)	84 (72–102)	<0.001
SBP, mmHg	135 (120–158)	131 (114–160)	0.2
DBP, mmHg	77 (66–88)	70 (60–84)	<0.001
STEMI	136 (37%)	28 (11%)	<0.001
Admission to ICU	362 (100%)	141 (55%)	<0.001
**Biological data**			
Troponin Ic (peak), µg/L	8.2 (1.8–36.0)	0.80 (0.2–5.3)	<0.001
CRP > 3 mg/L	215 (62%)	218 (88%)	<0.001
Hemoglobin, g/dL	13.8 (12.3–15.2)	12.1 (10.3–13.5)	<0.001
Creatinine, µmol/L	89 (72–114)	99 (77–146)	<0.001
**Angiographic data**			
Coronary angiography	351 (96%)	105 (41%)	<0.001
Non obstructive/normal	19 (5%)	16 (15%)	<0.001
Three-vessel disease	150 (43%)	59 (56%)	0.01
SYNTAX score	12 (5–21)	12 (2–23)	0.9
**Acute management**			
PCI	235 (64%)	29 (11%)	<0.001
CABG	29 (8%)	4 (4%)	0.1
Thrombolysis	13 (4%)	0 (0%)	0.05
**In-hospital mortality**			
Death	24 (7%)	38 (15%)	0.001
Death, CV causes	23 (6%)	24 (9%)	0.1

CABG: coronary artery bypass surgery, CAD: coronary artery disease, CRP: C reactive protein, CV: cardiovascular; DBP: diastolic blood pressure, HR: heart rate, ICU: cardiac intensive care unit, IQR: interquartile range, PCI: per-cutaneous transluminal intervention, SBP: systolic blood pressure, T1MI: type 1 myocardial infarction, T2MI: type 2 myocardial infarction.

**Table 3 jcm-08-02100-t003:** Logistic regression analysis of factors associated with type 2 myocardial infarction in patients with coronary artery disease (*n* = 619).

	Univariate		Multivariate	
	OR (95% CI)	*p*	OR (95% CI)	*p*
Obesity	0.33 (0.21–0.52)	<0.001	0.39 (0.23–0.66)	<0.001
Current smoking	0.33 (0.24–0.46)	<0.001	0.41 (0.27–0.62)	<0.001
Family history of CAD	0.41 (0.27–0.61)	<0.001	0.51 (0.32–0.83)	0.007
Heart failure	4.05 (2.62–6.26)	<0.001	2.98 (1.73–5.14)	<0.001
Troponin Ic	0.96 (0.95–0.98)	<0.001	0.97 (0.96–0.99)	<0.001
CRP > 3 mg/L	4.38 (2.84–6.76)	<0.001	3.53 (2.17–5.75)	<0.001
Acute renal failure	1.94 (1.40–2.68)	<0.001	1.22 (0.80–1.85)	0.4
STEMI	0.21 (0.13–0.33)	<0.001	0.37 (0.21–0.64)	<0.001

CAD: coronary artery disease, CI: confidence interval, CRP: C reactive protein, STEMI: ST-segment elevation myocardial infarction, OR: odds ratio.

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
