# Peer review of "Type 1 or Type 2 Myocardial Infarction in Patients with a History of Coronary Artery Disease: Data from the Emergency Department"

_jcm, 2019, doi:10.3390/jcm8122100_

Round 1

Reviewer 1 Report

I enthusiastically review this work by Putot en at. The authors used a database of 2830 patients admitted for MI in the ED to define whether a history of CAD predisposes to T1MI or T2MI and to determine distinctive characteristics of T2MI versus T1MI in the studied group; results disclosed that CAD is an independent predictor of T2MI when compared with T1M. Some methodologic comments to consider by the authors:

Abstract: please adjust to reflect acute atherothrombosis. Please soften the statement “but has not been evaluated in patients with previously known coronary artery disease (CAD).” To reflect it to not been evaluated extensively. Please refer to the data source in the abstract, and explain if the patients are consecutively included vs not.

Introduction: no additional comments.

Methods: Please include the country for this setting and clarify if the patients are consecutive or not. Also, clarify if the data was obtained electronically or by manual extraction, for these was any validation conducted for clinical characteristics? (beyond MI classification).  How was missing data handled? If just excluded a supplementary table comparing exclusions (i.e outlined in fig 1). Rationale for interaction testing needs to be included in the statistical section for the ones outlined in page 4, I do not see sex, this should be included as part of the interactions testing, please clarify.

The title for figure 2 needs to be adjusted, cannot see it in my version. Also please add positive/negative likelihood ratios and accuracy here.

Discussion: well written. Limitations please comment briefly on the external validity beyond France and Europe and other Race/ethnic groups.  

This study is strengthened by its sample size; the paper was well written and very easy to follow.

The study has a major contribution showing that CRP/troponin ratio could be of great help for T1MI / T2MI distinction in patients admitted for MI in the ED.

Author Response

Reviewer 1:

I enthusiastically review this work by Putot en at. The authors used a database of 2830 patients admitted for MI in the ED to define whether a history of CAD predisposes to T1MI or T2MI and to determine distinctive characteristics of T2MI versus T1MI in the studied group; results disclosed that CAD is an independent predictor of T2MI when compared with T1M. Some methodologic comments to consider by the authors: 

Abstract:

Please adjust to reflect acute atherothrombosis.

 Please soften the statement “but has not been evaluated in patients with previously known coronary artery disease (CAD).” To reflect it to not been evaluated extensively.

Please refer to the data source in the abstract, and explain if the patients are consecutively included vs not.

 We thank the reviewer for these remarks. Suggested adjustments and precisions have been added to the new manuscript:

Page 1 line 15 : “without acute atherothrombosis”

Page 1 line 16 : “has not been extensively evaluated in patients…”

Page 1 line 18 : Among 33,669 consecutive patients…”

Page 1 line 19 : 2830 patients with T1MI or T2MI were systematically included after prospective adjudication by the attending clinician, according to the universal definition.

Introduction: no additional comments.

Methods: Please include the country for this setting and clarify if the patients are consecutive or not.

The requested precisions are now available in the manuscript:

Page 2 line 63 : “ University Hospital of Dijon, France”

Page 2 line 61 : “all consecutive patients”

Also, clarify if the data was obtained electronically or by manual extraction, for these was any validation conducted for clinical characteristics? (beyond MI classification). 

As suggested, we added the following sentence in the new manuscript :

Page 3 line 105 : “Data were obtained by manual extraction from medical records for clinical data and electronically extracted for biological parameters. An independent validation was retrospectively performed for the clinical characteristics.”

How was missing data handled? If just excluded a supplementary table comparing exclusions (i.e outlined in fig 1).

We thank the reviewer for this relevant remark. Given the very low rate of missing data (0.7%), patients with missing data have been excluded from the analysis. We added this information in the new manuscript :

Page 2, line 71 : “Patients (…) with missing data (0.7%) were excluded.”

Rationale for interaction testing needs to be included in the statistical section for the ones outlined in page 4, I do not see sex, this should be included as part of the interactions testing, please clarify.

As requested, a new sentence has been added in the statistical section, and sex has been included as part of the interactions testing in the results section.

Page 3 line 118 : “The interactions between variables with a potential relationship (i.e. age*CRP>3mg/L; troponin peak*STEMI; hypertension*chronic kidney disease; history of CAD*STEMI: sex*STEMI; sex*smoking) were tested and added to the multivariate model if significant.”

Page 4, line 144 : “Significant interactions were tested in the multivariate model ((…)sex*smoking; sex*STEMI), without changing the results of the model.”

The title for figure 2 needs to be adjusted, cannot see it in my version.

We apologize for this layout issue, and modified the new version adequately.

Also please add positive/negative likelihood ratios and accuracy here.

These relevant information requirements have been added in the new submitted version (figure 2).

Discussion: well written.

Limitations please comment briefly on the external validity beyond France and Europe and other Race/ethnic groups.  

As suggested, a new limitation has been added in the new manuscript :

Page 10, line 296 : “These results concern a French, predominantly Caucasian population, and external validity could thus be limited in other populations.”

This study is strengthened by its sample size; the paper was well written and very easy to follow.

The study has a major contribution showing that CRP/troponin ratio could be of great help for T1MI / T2MI distinction in patients admitted for MI in the ED.

Reviewer 2 Report

Type 2 myocardial infarction (T2MI) is the result of an imbalance between oxygen supply and demand, without atherothrombosis. This study is based on 2380 MI patient to investigate CAD role in T2MI and is the first to identify CAD as an independent risk factor for T2MI, which will provide the clue to screen CAD in T2MI patient。

The authors did great science but need to edit carefully to clean the typo, label and adjust the figures properly before submitting to a journal. For example, Figure 2. The top panel covered part of lower figure.

Author Response

Reviewer 2:

Type 2 myocardial infarction (T2MI) is the result of an imbalance between oxygen supply and demand, without atherothrombosis. This study is based on 2380 MI patient to investigate CAD role in T2MI and is the first to identify CAD as an independent risk factor for T2MI, which will provide the clue to screen CAD in T2MI patient。

The authors did great science but need to edit carefully to clean the typo, label and adjust the figures properly before submitting to a journal. For example, Figure 2. The top panel covered part of lower figure.

We thank the reviewer for this relevant remark. The manuscript has been now carefully proofread and figure 2 has been readjusted.

Reviewer 3 Report

In this study, the authors evaluated incidence and characteristics of T2MI patients compared to T1MI. The distinction between T1MI and T2MI, as indicated by the authors, is crucial but difficult in practice. This is study provides new insights regarding treatment characterization, and a potential diagnostic to evaluate patients with CAD and T2MI in order to reduce costs and improve therapeutic strategies.

Major

In the abstract the authors state: In multivariable analysis, CAD history was an independent predictive factor 20 of T2MI versus T1MI (Odds ratio (95% Confidence Interval) = 1.40 (1.09-1.79)). Could the authors clarify where this data can be found? Please, include p-value. The authors should include the ROC curve statistics in the material and methods, statistical analysis section. Please, amend figure 2 since overlaps with the table. Line 174. I disagree that 41% of patients is a minority, please, rephrase. Line 173. point 3.2.3, section coronary angiography data. Based on the results of the study, 41% patients diagnosed with T2MI underwent coronary angiography of those 28% received PCI, meaning that 68% of patients undergoing coronary angiography received PCI. Interestingly, 96% of patients diagnosed with T1MI underwent coronary angiography of those 67% received PCI, meaning that 70% of patients undergoing coronary angiography received PCI. In the end, 68-70% of patients with MI (T1MI, T2MI) will undergo PCI (supported by the SYNTAX score, which is similar in both groups). Could the authors explain such finding? Line 183. No significant differences in cardiovascular deaths were observed, however, twice as much in-hospital mortality in T2MI was observed. How does the authors explain it? In my opinion, one explanation for such result is that the T2MI is an older population. I would suggest including a sentence in the discussion. The usefulness of the CRP/troponin ratio resides in the previous diagnosis of CAD, which limits the overall efficacy of this diagnostic tool for undiagnosed CAD patients at presentation. Please, include a phrase in the conclusions or limitations of the study section.

Minor

Line 19. I suggest including ‘fourth’ as for the Universal definition in the abstract Line 21 and Line 222. I would like to suggest the authors 1) consistency when presenting OR and HR values (e.g: ‘:’ vs. ‘=’). In the abstract, could the authors include the p-value as well. Line 27. Please include hyphen in C-statistic. Line 113. Since type 2 MI was already abbreviated, please include the abbreviation. Line 214. Remove the space after the full stop.

Author Response

Reviewer 3:

In this study, the authors evaluated incidence and characteristics of T2MI patients compared to T1MI. The distinction between T1MI and T2MI, as indicated by the authors, is crucial but difficult in practice. This is study provides new insights regarding treatment characterization, and a potential diagnostic to evaluate patients with CAD and T2MI in order to reduce costs and improve therapeutic strategies.

Major

In the abstract the authors state: In multivariable analysis, CAD history was an independent predictive factor of T2MI versus T1MI (Odds ratio (95% Confidence Interval) = 1.40 (1.09-1.79)). Could the authors clarify where this data can be found? Please, include p-value.

As relevantly noticed, Odds ratio in the abstract was wrong and has been modified in the new manuscript, according to the table 1. We apologize for this error.

Page 1 line 22 : “(Odds ratio (95% Confidence Interval) = 1.38 (1.08-1.77))”

The authors should include the ROC curve statistics in the material and methods, statistical analysis section.

As suggested, the ROC curve statistics have been adequately added in the statistical analysis section.

Page 3 line 125 : “To compare accuracy of biomarkers and to identify best cut-off values for T2MI prediction, we constructed receiver operating curves (ROC) and determined the area under the curves, as well as sensitivity, specificity, accuracy, and positive and negative likelihood ratios for each parameter.”

Please, amend figure 2 since overlaps with the table.

We apologize for this layout issue, and modified the new version of figure 2 adequately.

 Line 174. I disagree that 41% of patients is a minority, please, rephrase.

As requested, this sentence has been rephrased :

Page 8, line 174 : “Less than half (41%) of patients…”

Line 173. point 3.2.3, section coronary angiography data. Based on the results of the study, 41% patients diagnosed with T2MI underwent coronary angiography of those 28% received PCI, meaning that 68% of patients undergoing coronary angiography received PCI. Interestingly, 96% of patients diagnosed with T1MI underwent coronary angiography of those 67% received PCI, meaning that 70% of patients undergoing coronary angiography received PCI. In the end, 68-70% of patients with MI (T1MI, T2MI) will undergo PCI (supported by the SYNTAX score, which is similar in both groups). Could the authors explain such finding?

We thank the reviewer for this relevant remark. There was indeed an error in the presented proportions, as only 29 T2MI patients (i.e. 11% instead of 28%) had PCI. Similarly in the T1MI group, only 235 patients (i.e. 64% instead of 67%) had PCI.  The proportion of patients receiving PCI among patients undergoing coronary angiography is 28% for the T2MI group and 67% for the T1MI group.

We modified the percentages in the table 2 and clarified the text :

Page 6, line 173 : “Among patients who had coronary angiography, only 28% of T2MI cases had a PCI, compared with 67% of T1MI patients.”

Line 183. No significant differences in cardiovascular deaths were observed, however, twice as much in-hospital mortality in T2MI was observed. How does the authors explain it? In my opinion, one explanation for such result is that the T2MI is an older population. I would suggest including a sentence in the discussion.

We fully agree with the reviewer that older age probably explains twice-higher all cause in-hospital mortality after T2MI, although no significant difference was observed for cardiovascular mortality.

This relevant remark has been added in the new manuscript :

Page 9 line 263 : “Interestingly, when compared with T1MI, T2MI was associated with twice higher in-hospital mortality, but no significant difference in cardiovascular deaths. These results suggest an excess risk of non-cardiovascular events in the T2MI population, who were 10 years older than T1MI patients, and for whom iatrogenic risk of invasive procedures should be carefully weighted.” Conversely, older patients with T1MI should not be excluded from PCI, as PCI has been associated with substantial improvements in health-related quality of life, similar to those in younger patients [23]”

The usefulness of the CRP/troponin ratio resides in the previous diagnosis of CAD, which limits the overall efficacy of this diagnostic tool for undiagnosed CAD patients at presentation. Please, include a phrase in the conclusions or limitations of the study section.

We added this relevant limitation in the new manuscript :

Page 10, line 300. “Finally, the usefulness of the CRP/troponin ratio has been evaluated in patients with previous diagnosis of CAD, which limits the overall efficacy of this diagnostic tool for undiagnosed patients at presentation.”

Minor

Line 19. I suggest including ‘fourth’ as for the Universal definition in the abstract Line 21 and Line 222. I would like to suggest the authors 1) consistency when presenting OR and HR values (e.g: ‘:’ vs. ‘=’). In the abstract, could the authors include the p-value as well. Line 27. Please include hyphen in C-statistic. Line 113. Since type 2 MI was already abbreviated, please include the abbreviation. Line 214. Remove the space after the full stop.

We thank the reviewer for these suggested corrections that have been implemented in the new manuscript, except for the Suggestion on the line 19.1, because our study was performed before the fourth Universal definition. Thus, we have not added ‘fourth’ in the abstract to avoid anachronism, even if T1MI/T2MI distinction criteria did not substantially differ between the third and fourth Universal definitions.